# The Emirati Vernacular: Tracing the UAE's Art History through Architecture as a Reflection of Society

Eve Grinstead 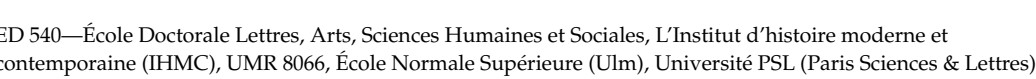

ED 540—École Doctorale Lettres, Arts, Sciences Humaines et Sociales, L'Institut d'histoire moderne et contemporaine (IHMC), UMR 8066, École Normale Supérieure (Ulm), Université PSL (Paris Sciences & Lettres), 75005 Paris, France; evegrinstead@gmail.com

**Abstract:** While past studies have considered the relationship between art and architecture, art and society, or society and architecture, few consider all three, let alone when discussing these subjects in the United Arab Emirates. This article presents the evolution of the art scene in that country's three largest emirates, from its foundation as a federation through today, as a reflection of local societal truths. Since its early days, each concerned emirate—Abu Dhabi, Dubai, and Sharjah, has developed an art scene unique from the others, and each one has been housed in different kinds of mostly vernacular—though sometimes academic—architecture. Through data collection of the various types of architecture employed in each emirate, this article explores possible reasons why each state has its own art scene, and what can explain this phenomenon. Abu Dhabi, the wealthy capital, has "starchitect" designed institutions; the more avant-garde Dubai employs recycled industrial hangars (or structures made to look as such); while the more traditional Sharjah repurposes historical structures for artistic use. Over time, each emirate begins to borrow different architectural tendencies hitherto mostly seen in the other states, demonstrative of the constant competition the three emirates have with each other. Beyond the local implications, these structures provide a rich discussion on what is considered vernacular in a modern context, as well as where the definition of one stops and the other begins.

**Keywords:** Emirati art scene; manifestations of power through art and architecture; vehicular vs. vernacular space; grassroots art scenes; women in the arts; royal families in the arts; arts in the Gulf

## 1. Introduction

The Emirati art scene provides ample fodder for a discussion that reevaluates unrecognized expressions of vernacular architecture; furthermore, the evolution of this scene is a good case study to demonstrate the relationship between art, architecture, and society. In a span of just fifty years, starting with independence from the British to the present day, the local architectural landscape has metamorphosed from instances of ephemeral Bedouin structures, clusters of wind towers or other traditional constructions surrounding commercial hubs, and the occasional industrial zone, to soaring skyscrapers, starchitect designed institutions, and sprawling suburban sectors that would indicate we are anywhere but the Arabian desert. Today, there is a lack of architectural coherence as the never-ending construction of newer buildings foreshadowing the future contrasts with the country's nomadic past. Dissecting and analyzing the local architectural transformation that has evolved in the last half-century is far too complex for the scope of one article. We can, however, take a detailed look at the growth of the local art scene through its architecture and discuss how changes in both reflect Emirati society. This tripartite thought process can shed light on other aspects of Emirati architecture and give us further reason to consider further use of such complex ways of comparing local art, architecture, and society.

As a brief history, the UAE became a federation of seven states (see Figure 1)—Abu Dhabi, Dubai, Sharjah, Umm al-Quwain, Fujairah, Ajman, and Ras al-Khaimah—following

the British military's withdrawal from the region. After nearly 150 years as the Trucial States, the United Arab Emirates formed in 1971. Mindful that having a scant population, and hitherto depending on the British for the development of infrastructure, the Federation's founder and "Father", Sheikh Zayed encouraged the idea of perpetuating the employment of foreigners for the country's development. He also, with the help of his wife Sheikha Fatima, as an emissary figure from the royal family to all women of the newly founded nation, promoted the idea of women taking a greater role in public society and the workplace to help the country advance. An unplanned result of this years later was the genesis of a local art scene.

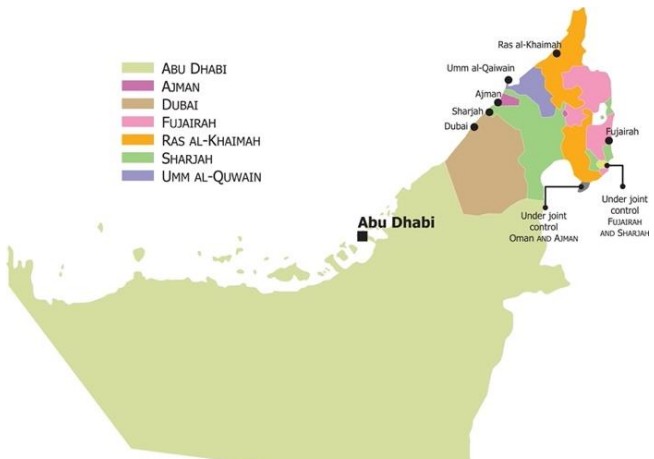

**Figure 1.** Map of the seven states of the United Arab Emirates. Used with permission from Pngegg.com, accessed on 1 June 2021.

To normalize the idea that women were essential for the new country's survival, the "Mother of the Nation" established the Woman's Union of Abu Dhabi to teach women to read and write Arabic, as well as other languages, among other skills. Soon after, other royal women opened similar organizations in their respective emirates, and since then, it has been critical for women of similar pedigree to start their philanthropic initiatives.

Moving forward a few decades, this tendency evolved among sheikhas to establish art centers (or another project to support the arts and artists). That these royal women endeavored to create something untraditional and non-existent for earlier generations, following in the footsteps of their foremothers—and that their families accepted those undertakings—set a precedent for other women to do so. This "approval" of women leading in the arts catalyzed the process for other locally based, but non-Emirati, women (whether UAE-born or not) to initiate similar artistic projects of their own.

While the contribution of local women to what has become the art scene in the UAE cannot be disregarded, some significant structures established by foreign women remain essential to the growth of an art scene in the country's three largest emirates: Abu Dhabi (the capital), Dubai, and Sharjah. As with other federations, each state varies slightly in terms of society, culture, and policies. This concept holds for the arts.

Various examples of architecture in each emirate express these discrepancies, in addition to intra-emirate rivalry and influences. After the country's establishment in 1971, the art community emerged only a few years later, in the late 1970s and early 1980s, through various structures, both commercial and not-for-profit sectors. In 1979, Sheikh Dr. Sultan bin Muhammad Al-Qasimi, the ruler of Sharjah, declared a "Revolution of Culture" for his emirate, which emphasized the fine arts and higher education, established a book fair, and a theatre (Kazerouni 2017). That same year, in Dubai, British expatriate Allison Collins founded the country's first art gallery—Majlis—in her home in Bastakiya (Collins 2020). The Emirates Fine Arts Society (EFAS) opened in 1980 in Sharjah to host classes and annual exhibitions for its members and invited artists. A year later, on the impetus of Sheikh

Zayed, the Cultural Foundation in Abu Dhabi opened as the home of an arts and culture center, including the first national library.

Thus, from the country's nascent days, the three emirates in question have been host to a growing art scene. How each example was started, upon whose initiative, in what kind of structure, and for what reasons varies greatly. Examples of each emirate's particular "flavor" have come to exist in the others, demonstrating the states' deeply rooted competition, different societal values, as well as different "shades" of the Emirati art scene, but also their influence on one another. We tend, in the West, to blur the nuances between these states; however, it is necessary to take a detailed look at the origins of each art scene. Beyond the local implications, these structures provide a rich discussion on vehicular vs. vernacular architecture. Employing an approach that employs Soja's notion of the Thirdspace is a useful technique to reveal truths about Emirati society through manifestations of art and architecture.

Through this article, we will go into detail on the three different art scenes present (in Abu Dhabi, Dubai, and Sharjah), and the main characters of each one. The author has employed a data collection method to list as many instances and examples of the local art scene in each state as possible, what kind of architecture has been employed the most, and who, or what, is behind these structures.

For each state, this discussion will include what kind of architecture is employed to house each manifestation of the art scene, what kind of building it is (repurposed or purpose-built, traditional or modern), who designed it (local, unknown architects or world renown starchitects), and who is behind the evolution of the art scene in that emirate (a royal family, private individuals, a business, a mix of both or all three).

Then, because each state has characteristics that are more associated there than the other two (to be detailed in Section 3), and because the three states are in constant competition to surpass the others, as the art scene has evolved from the early days, each state has tended to have examples of architecture for their art scene that seem to have been inspired from the other two states (Section 4).

Following this cross-analysis, the article explores this complex example of art and architecture as a reflection of society in a larger context of architectural history and analysis. For this component, we apply the use of terms such as vehicular or academic architecture and that of the vernacular, but to the case of Emirati architecture that is not nearly as old as some of the traditional examples of the term. Many of the buildings employed in the art scene in the UAE are unrecognized expressions of vernacular architecture as they stray from the traditional definition of the term.

The article puts forth, after the initial comprehensive description of each emirate's art scene, that the architecture used to house each scene, different in each state, reflects each emirate's society. Beyond that, by discussing how each scene interacts and competes with the others, this rivalry is also reflected through more recent examples of architecture used in the three art scenes that had previously only been used elsewhere.

## 2. Literature Review

While the topic of tracing art history through architecture is not unique, doing so to analyze society is a less traveled road. Furthermore, given the relatively short history of the UAE (founded in 1971) and its art scene (beginning in the mid-to-late 1970s), studying the art scene of the UAE through its architecture has not been as explored as that of Western counterparts. Existing literature that does cover some of the topics discussed in this article tends to be either geographically too broad, or too specific, and only covers a few of the issues we touch upon. Publications on the country's architecture, or that of the three states discussed here, include *Lest We Forget: Structures of Memory* (Bambling et al. 2014), from the UAE's inaugural participation in the Venice Biennial or *Transformations: The Emirati National House* from the 2016 Pavilion (Elsheshtawy and Aravena 2016), and *Lifescapes Beyond Bigness* from the 2018 iteration (Alawadi 2018), consider various types of Emirati architecture and how they have evolved with the modernization of the country and society

changes; however, none of them addresses the use of architecture and its role in the local art scene(s). Frauke Heard-Bey's publication *Abu Dhabi, the United Arab Emirates and the Gulf Region. Fifty Years of Transformation* includes distinct chapters on architecture ("Changing Adaptations—Adapting to Change: The Architecture of the UAE A Historical Background to Traditional and Modern Living Conditions in the UAE") and art ("Art in the United Arab Emirates"), but not a separate chapter on how the two have interacted or what either's relationship is with local society (Heard-Bey 2017).

Other publications go into greater detail on one specific Emirate's architecture and how its relationship to society. For instance, *Building Sharjah* (Al Qassemi and Reisz 2021), *Showpiece City: How Architecture Made Dubai* (Reisz 2021), or *Abu Dhabi Public Spaces: Urban Encounters, Social Diversity and (In)Formality* (Kyriazis et al. 2021) delve into details on one emirate's architecture and how it reflects local society, or vice versa; but again, the publications are limited to one state, and none include an art-scene specific angle.

Covering a far vaster geographical area, several publications discuss "Arab Architecture" and how it can be a reflection of society, identity, politics, or nationalism, but by discussing such a large region, the specifics of Emirati architecture in its art scenes are left out (For example: (Isenstadt and Rizvi 2008; Andraos et al. 2016; Arbid and Oswalt 2022)).

On the other hand, *World Culture Districts: Spaces of the 21st Century*, (Strasser et al. 2021), includes a chapter on Dubai's Alserkal Avenue gallery hub, addressed later in this article, but the book also covers similar districts around the world and does not include other areas in the UAE. Also more specific than the tripartite comparison in this article allows, *Re-Using Heritage Elements in New Buildings: Cases from the United Arab Emirates* by Jihad Awad and Bouzid Boudif (Awad and Boudiaf 2020), discusses the increasing use and re-use of vernacular architecture in the UAE; the focus; however, being more on specific architectural elements, this article neither goes into great detail on the relationship of this architecture to society, nor does it discuss at all the local art scene.

## 3. Materials and Methods

This article is the result of a qualitative approach based on the investigation and analysis of various buildings that house examples of the art scene in the three largest states of the United Arab Emirates.

## 4. Results: The Different Local Flavors and Origins of Emirati Art and Architecture

Three of the seven emirates host an art scene that is at the same time unique from the others, and yet also benefits from the differences that the neighboring states provide and, to some extent, mimic them. Depending on what we choose to elect as the most imperative factor in defining an art scene will determine how to present and classify the three different art scenes.

Today, Abu Dhabi, the federation's wealthy capital, is most known for its "starchitect" designed institutions, often conceived as a local branch of a long-existing foreign establishment. Not surprisingly, being a commercial hub, Dubai is reputed for its annual art fair, Art Dubai—the largest in the region—in addition to locally based grassroots commercial galleries. As for Sharjah, its biennial has long attracted art lovers, and it is also the core from which the Sharjah Art Foundation (and all its derivatives) have sprung. Nearly all art initiatives come from either the current ruler, Sheikh Sultan bin Muhammad Al-Qasimi, or his daughter, Sheikha Hoor Al-Qasimi.

The above paragraph on the three "artistic" emirates gives a succinct description of each state's scene; it does not present the same information for each state. Table 1 provides a summary of similar aspects for each scene, into which we will further delve in later paragraphs to give a more in-depth description and context. We also explain how each emirate, however diverse, benefits from the differences presented by the others and how these differences also inspire the others to create new developments in their state. We will go through the three emirates in alphabetical order, also moving geographically from West to East: Abu Dhabi, Dubai, and Sharjah.

**Table 1.** Comparative table of the art scene in Abu Dhabi, Dubai, and Sharjah.

| Emirate | Abu Dhabi | Dubai | Sharjah |
|---|---|---|---|
| Architecture of art scene | Mostly purpose-built/starchitect designed, as a local branch of a foreign institution | Warehouses primarily. Some historical houses. Previously also in hotels, malls, towers. Few new/purpose-built structures | Mostly preserved historical houses or refurbished industrial spaces. Few new/purpose-built structures |
| General trend of founders or Patrons | Government entities supported by the royal family | Galleries = mostly expat women. Fair = group of expats (m/f mix) | Al-Qasimi family (Sheikh Sultan or his daughter Hoor) |
| Financial Structure | Large scale institution non-profit (heavy government funding/subsidy) | Galleries are small businesses with usually one branch. Small foundations or residences are non-profit. More recent museums vary between state funded and individual wealthy family trusts | Exclusively non-profit |
| International Partnerships | Many | Some | Occasionally/in some instances |
| Role of Royal Family | A few select members of local royal family (Al-Nahyan) found or head committees that have contributed to the creation of institutions | Usually as sponsor, less common as founder/director (Al-Maktoum) | Heavy: royal family financially supports institutions and important family members found/run the institutions (Al-Qasimi) |

### 4.1. Abu Dhabi

In terms of its art scene, Abu Dhabi today is known perhaps most famously for the Louvre Abu Dhabi, on Saadiyat Island (see Figure 2), announced in 2007 but only completed in 2017. Theoretically, the local branch of the most visited museum in the world (Da Silva 2022) is not unique to this 27 km$^2$ island; there are also plans to host a local Guggenheim, among other museums.

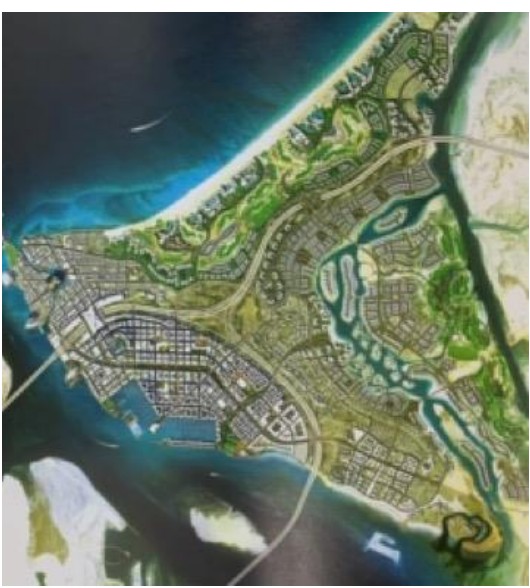

**Figure 2.** Aerial view of the proposed construction on Saadiyat Island. From *Saadiyat Island Cultural District Exhibition* brochure (undated). Photograph by the author from the Jameel Art Centre Archives.

In March 2007, an agreement was signed between Agence France-Muséums (or AFM) and Abu Dhabi Tourism Authority (or ADTA), announcing the creation of the Louvre Abu

Dhabi. The group revealed that Jean Nouvel would design this new museum, and among its neighbors was to be the next iteration of the Guggenheim network, designed by another starchitect: Franck Gehry.

While the Louvre opened after several delays in 2017, local economic woes in addition to continued protests of the emirate's human rights practices delayed the groundbreaking of Gehry's next Guggenheim. In 2021, the Guggenheim brand revealed that construction should soon begin and that the opening would take place in 2026 (Hilburg 2021).

Even though the newest Guggenheim iteration is not yet open, and Louvre did not do so until 2017, both museums have long had a presence in the local and international art community as if they have always been open. Since 2010, these institutions have appeared in local art guides (for example in *Art Map* (2010), starting in issue 11), and soon after each began hosting events in off-site locations such as their Talking Art Series. Beyond the art scene, but significant to note in a discussion on Abu Dhabi's architecture and society, several other museums fall into this trend of illustrious institutions designed by world-class architects.

Whereas they have little in common in terms of their internal contents or mission, these museums share geographic proximity in that they were all designed for Saadiyat Island. The ADTA bought this natural island located slightly northeast of Abu Dhabi in 2004 to develop it into a "signature destination with environmentally sensitive philosophies" (Tourism Development & Investment Company n.d.). The Abu Dhabi government created the ADTA for that specific purpose and later changed its name to the Abu Dhabi Tourism and Culture Authority (or the ADTCA) (Kazerouni 2017). In 2012, the Abu Dhabi Department of Culture & Tourism launched, joining the Abu Dhabi Authority for Culture and Heritage, Abu Dhabi Tourism Authority, and the Tourism Development & Investment Company (abbreviated as the TDIC), a private development company. Regardless of the multiple name changes, the overall purpose has always been to promote culture and tourism in Abu Dhabi, and they have always been largely government entities or private ones created with those specific missions in mind.

Among the original structures destined for "the Island of Happiness", in addition to the Louvre and the Guggenheim, is the Sheikh Zayed National Museum, a performing arts center, and a Maritime Museum—alluding to the country's past in pearling and fishing, were planned (see Figure 3). It is noteworthy, yet no wonder, that the architects for these three structures were all Pritzker-prize winning: Norman Foster, Zaha Hadid, and Tadao Ando, respectively. To this day, the only one planned to open is the Norman Foster's Zayed National Museum. Abu Dhabi's Department of Culture and Tourism website refers to neither Hadid's "Fruits on the Vine" performing arts center nor Tadao Ando's dhow (a traditional local boat) shaped maritime museum. The website still discusses the Zayed National Museum in the future tense but gives no opening date. The British Museum, originally a partner of the Zayed National Museum, renounced its collaboration in 2017, but currently, the structure still seems to be planned for construction later.

Art and culture in Abu Dhabi are not limited to these, longtime mostly theoretical, museums, but they are perhaps most emblematic of the emirate's "flavor" of art and architecture. They are also reflective of the country's perpetual future thinking. While less well known internationally, the Cultural Foundation, which opened on Sheikh Zayed's initiative in 1981, can be seen as the precursor to the Abu Dhabi versions of the Louvre, the Guggenheim, and the other starchitect-designed museums. The Abu Dhabi government launched the competition for the Cultural Foundation in the 1970s, nearly a decade before its opening.

Founded by the world-famous Bauhaus architect Walter Gropius, the Architects' Collective's (or TAC) proposal won the competition and finalized their plans by 1975. From its inception, Abu Dhabi thus has had a tradition of conceiving large-scale institutions designed by world-renowned foreign architects.

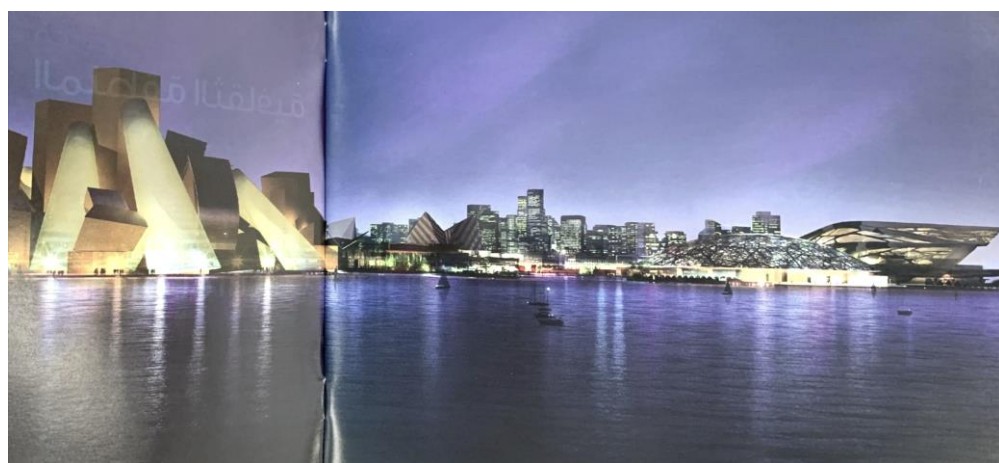

**Figure 3.** Mockup of the proposed museums on Saadiyat Island. From *Saadiyat Island Cultural District Exhibition* brochure (undated). Photograph by the author from the Jameel Art Centre Archives.

If the Cultural Foundation is home to the first national library, performance hall, and cultural center—and illustrative of the emirate's and country's onward perspective to the future—there are also traditional structures that were "sanctified" even before this one, historical structures that look onto the past. For example, the government inaugurated the Al Ain Palace Museum (Abu Dhabi's smaller city and Sheikh Zayed's birthplace) in 1971. While the Qasr Al Hosn Museum only opened in 2018, this historical fortress tower is considered one of the most important architectural monuments of the Emirate. It is noteworthy that the government chose to build the Cultural Foundation just across from it to bring the future and the past together in the new country. This juxtaposition also brings together examples of both academic and vernacular architecture.

One can observe this contrast—academic vs. vernacular—elsewhere in Abu Dhabi as well as in the two other emirates; however, the juxtaposition is not so striking in the other cases. Beyond that, this contrast is present in many levels of Emirati society, not as a visual distinction, but rather as a nearly omnipresent contradiction between a nostalgia for a quickly fleeting past and a coveted yet overwhelming onset of the future. We shall explore different examples in the other emirates before comparing the three together.

### 4.2. Dubai

If Abu Dhabi's artistic presence is mostly defined by large institutions that are the products of international agreements—suitable for a capital city and the wealthiest emirate—it is no surprise that the artistic landscape of the more commercial Dubai has been composed more historically with more commercial structures. In Dubai's case, the presence of galleries makes up the local art scene. Individually smaller and less well known than the larger institutions in Abu Dhabi—and later Sharjah—together, these grassroots structures characterize the art scene in Dubai. The first gallery was opened by British ex-pat Allison Collins in 1979—in her home in the Bastakia/Al Fahidi district, i.e., the "Old Dubai" district that covers approximately 0.06 km$^2$ of the now sprawling city (see Figure 4). It was only in the 1990s when other galleries began to spring up with five galleries opening between 1995 and 1998; at this time, Abu Dhabi only had one, and Sharjah had none. Twenty-one more galleries launched between 2003 and 2009, and forty-nine between 2010 and 2020.

Dubai's first galleries opened either in the owner's home or, slightly later, in warehouses in the larger and more industrial district Al Quoz—about the same size as Saadiyat island. These structures reflect both the city's more distant (merchant houses in Old Dubai) and more recent (warehouses to serve the ports of Sheikh Saeed and Jebel Ali) past. For the former, Allison Collins opened the Majlis Gallery in her home in Old Dubai for the simplicity of doing so nearby. This area is now referred to as the Al Fahidi historical district and hosts many examples of houses owned by pearl merchants or other wealthy traders

whose business was in proximity to the Khor Dubai (Dubai Creek). Today, it is in the literal and figurative shadows of buildings in Downtown Dubai (Dubai Mall, Burj Khalifa, Dubai International Financial Centre, and more recently, the Museum of the Future, and the Dubai Frame), the Palm, Dubai Marina, and other developments. At the time of Collins' arrival in Dubai, many of these areas did not yet exist, and she found the ancient quality of these crumbling buildings charming. Several years later, in 2003, Mona Hauser also opened a gallery and hotel in this area: the XVA Gallery.

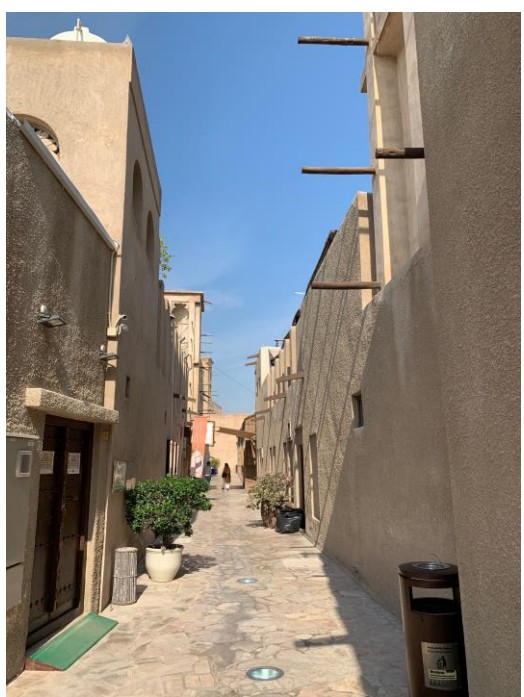

**Figure 4.** A typical street in the Bastakia/Al Fahidi District. Photograph by the author.

While she did not live there, she describes appreciating a similar charm to these buildings, especially in contrast with all the new construction. More importantly, that she also chose to recycle an existing structure built for a specific purpose other than being an art gallery is remarkable. In 1995, the Green Art Gallery, started by the Atassi sisters as the Ornina Gallery in Homs, Syria, opened in another residence in Dubai: a villa in the beachfront neighborhood of Jumeriah. Again, the main reason to open chez soi was out of simplicity to not have to rent a new location.

Beginning in the mid-1990s, warehouses, mostly in the Al Quoz industrial district, became the next kind of structure used to house galleries. The first example of this is the Courtyard/Total Arts Gallery. Today, many of the most famous galleries in Dubai are in this area. Like other examples of transforming former industrial space into exhibition locations, the lower cost of rent in these less-desirable locales motivated gallerists to slowly move there as rents in other parts of the city kept creeping up. Reflective of trends elsewhere in the art world of artists recycling factories, hangars, and other manufacturing edifices, the use of these structures also reflects trends in Dubai's history.

Later, as the art scene grew, galleries began to rent spaces in more modern structures— not residences or warehouses as we have seen in the past, but constructions catering to, and demonstrative of, a growing transient population such as hotels, malls, and retail floors of the city's numerous skyscrapers. Well-reputed galleries such as Tabari Artspace, 1x1 Gallery, Andakulova Gallery, East Wing (now closed), Opera Gallery, and Lumas, although in different locations today, are some examples. Other less well-known examples include Gallery One, Legacy Art, Sovereign Gallery, Monde Art, Profile Gallery, DUCTAC (nonprofit), and the Vindemia Gallery. This trend is reflective of the fact that there was indeed a growing art scene, but not specific neighborhoods associated with the arts, ex-

plaining why, in the early days, galleries existed throughout the city in structures that seem like a curious choice for an art gallery.

The use of malls or hotels as a host for galleries is unusual in the West; however, it should rather be thought of as a gallery in a souk or bazaar. Such markets have a long tradition in the Middle East for being hubs of commercial activity. In the UAE, and Dubai especially, malls continue this tradition, and the fact that they are air-conditioned also helps to attract an audience and foot traffic not found elsewhere. As for hotels, these constructions often lend themselves as temporary hosts of conventions or other meetings. With this purpose in mind, some of the original gallery owners rented space in hotels, knowing that an international audience would see their exhibited artwork. Finally, when navigating Dubai, one rarely refers to streets when giving directions, save for large thoroughfares such as Sheikh Zayed Road, Al Wasl Road, Al Manara Street, etc., but rather by tower, hotel, or mall names. With the rapid rate of construction in this city, roads can quickly appear or disappear as urbanization continues; it makes more sense to navigate using structures that people can see and that are known.

Until 2007–2008, galleries existed in locations built for other purposes: residential (old or new), recycled industrial warehouses, and non-specific retail space. This year is a crucial moment for the art scene in Dubai (as well as elsewhere in the UAE) as it marks the start of a new era with the officialization of Alserkal Avenue, the DIFC Gulf Art Fair (later branded as Art Dubai—today the region's biggest art fair), and the first local sale from Christie's auction house. In 2008, the DIFC's (Dubai International Financial Centre) Gate Village opened as a commercial and retail hub to service this increasingly important business free zone. While the Gate Village is only 0.02 km$^2$, the entire DIFC spreads over 0.56 km$^2$. Conceived to host luxury shops and lavish restaurants, several previously established galleries moved there, while other new galleries opened.

Among the former, these include the Tabari Art Space (moving from the Fairmont Hotel Downtown), and the XVA Gallery, which opened a second location there for a few years, while new galleries such as Cuadro, Art Sawa, Opera Gallery, and the Empty Quarter all started there around this time. Christie's also had their offices there, while the first version of Art Dubai took place in the DIFC before moving to Madinat Jumeirah, though its offices remained there until moving to Dubai Design District years later. In this case, we are getting closer to purpose-built architecture for art exhibitions, whereas in the past, gallerists would reappropriate vernacular architecture previously built for an entirely different use. The DIFC Gate Village was conceived with a more specific purpose than retail space in malls, hotels, and skyscrapers, but was, nonetheless, not designed for art exhibitions.

In 2013, Dubai Design District (abbreviated as d3) was announced as a 0.08 km$^2$ planned community to host businesses and organizations whose work revolves around design, art, fashion, etc. In the larger scope of Dubai's urbanization, it is not unlike other real estate developments in Dubai: owned and constructed by TECOM investments. This group is a branch of Dubai Holding; Dubai's ruler, Sheikh Mohammed bin Rashid al-Maktoum owns most shares of this company. D3 is not the first kind of development for this group that designates a discipline or subject to a new part of the city; these include Dubai Internet City, Dubai Outsource City, Dubai Media City, Dubai Studio City, Dubai Production City, Dubai Knowledge Park, Dubai International Academic City, Dubai Science Park, and Dubai Industrial City. Concerning just the arts, however, this is the only example where the local government created buildings, let alone an entire neighborhood, for the arts. The artistic specification is noteworthy because, unlike Abu Dhabi and Sharjah, early milestones in the Dubai art scene were made by civilians and not by government entities or members of the royal family.

Indeed, the cluster of galleries that became more officially known as "Alserkal Avenue" (see Figure 5) in 2008 started out as just that: a few rows—or avenues—of warehouses that previously only had industrial purposes—some are still there to this day. Adjacent to these structures stood a marble factory that the Alserkal Avenue group tore down in 2012 to

make way for a mirroring expansion that would open in early 2015. In this case, Alserkal Avenue belongs to the wealthy Al Serkal Family.

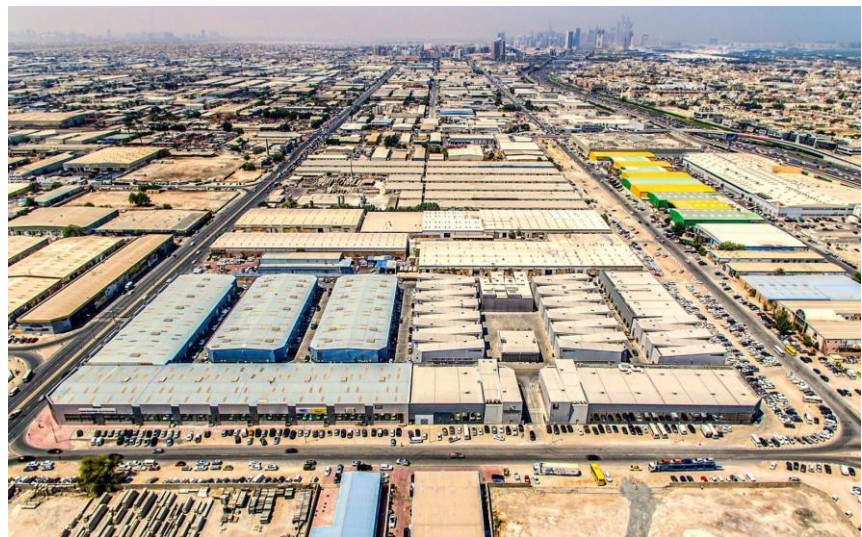

**Figure 5.** Aerial shot of Alserkal Avenue, 2017. The original warehouses are visible on the left and the newer expansion is on the right. Image courtesy of Alserkal Avenue.

This family, whose "Alserkal Group" was established in 1947, well before the federalization of the country, owns, or are the local shareholders of, such corporations as Bridgestone tires (since 1948), Pepsi (since 1962), the Commercial Bank of Dubai (since 1969), Beit Al Khair society (a philanthropic society since 1970), Etisalat (one of the country's two largest telecommunications company, since 1976), Dubai Insurance (since 1989), Dubai Electric and Water Authority (since 1992), Dubai Autism Center (since 2001), Al Mal Capital (since 2005), and Emirates NBD bank (previously the National Bank of Dubai, since 2007). Local but not royal, the Al Serkal family is nonetheless a key player in Dubai's overall development, including its art scene. While the whole family is involved, Abdelmonem Bin Eisa Alserkal, the grandson of the family's founder (Nasser bin Abdullatif Alserkal), is the family member most involved with this endeavor, along with his long-time collaborator, Lithuanian-born Vilma Jurkute.

The expansion doubled the size of the "Avenue", to about 0.05 km$^2$, and the new construction mirrored the original ones. The architects conceived the new buildings to resemble the original warehouses but with the idea to host new art spaces, among other purposes. Unlike Abu Dhabi, which always had a long history of employing internationally renowned architects, Alserkal Avenue opted for a local option: QHC Architects (based in Sharjah) and EDMAC Consulting (based in Dubai and Fujairah).

For the most part, the architecture that hosts Dubai's art scene are structures previously not used to display art but reappropriated to do so: residential villas, hotels, malls, towers, and warehouses. Districts such as DIFC and d3 perpetuated and specified the use of hotels, malls, and towers. The new warehouses at Alserkal Avenue copied the aesthetic of its original warehouses (not built for art) for a more bespoke arts purpose. In Dubai, being that the early examples of the art scene came from individuals and not government initiatives, it makes sense that the original structures to house the art scene were simple, vernacular examples of places that anyone could access i.e., not large new buildings as we have seen in Abu Dhabi (or later in Dubai with parts of the DIFC and d3) or reusing significant historical structures that are government owned. The latter method, we observe in Sharjah.

*4.3. Sharjah*

In the case of Sharjah, many of the buildings that house the art scene are historical—while such examples exist in Abu Dhabi and Dubai, that is the exception in those emirates.

Select members of Sharjah's ruling family (the Al-Qasimis) run the local art scene, which is mostly located in the center of the old town (approximately 0.035 km² also referred to as the "Heart of Sharjah" or the "Arts District, see Figure 6); the use of historical buildings is, thus, not surprising.

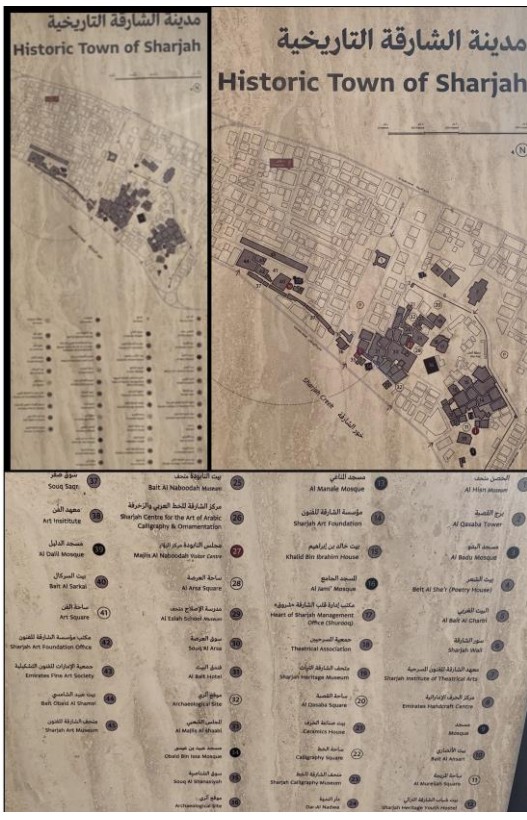

**Figure 6.** Panel (with details) indicating Sharjah's historic center. Photograph by the author.

While in Abu Dhabi the first example of architecture to display art and culture (the Cultural Foundation) is indeed in the historical center of the city, today it has shifted to Saadiyat Island, specifically designed by the government to host strongholds of culture. In Dubai, similarly, the first example of a gallery was along the Creek in the older part of the city, but the art scene has also migrated to newer neighborhoods, emblematic of its future: the industrial Al Quoz, the Dubai International Financial Centre, and in later years, Dubai Marina.

Today's art scene revolves around the Sharjah Art Foundation, presided by Sheikha Hoor Al-Qasimi, the current ruler's daughter. She established the SAF in 2009 as the permanent iteration of the Sharjah Biennale, founded in 1993 by her father Dr. Sheikh Sultan Al-Qasimi. He is solely responsible for creating the institutions in Sharjah that celebrate its culture (Sharjah Archaeology Museum, Sharjah Art Museum, Sharjah Calligraphy Museum, Sharjah Museum of Islamic Civilization, etc.), all under the auspices of the Sharjah Museums Authority. Sheikh Sultan has been promoting traditional culture and art in the larger sense of the term (theater, literature, calligraphy, etc.) since the 1980s. He was also the first leader to do so specifically for contemporary art: he launched the not-for-profit (a rare status at the time in the newly founded country) Emirates Fine Arts Society (or EFAS) in 1980 through the Ministry of Labor and Social Affairs. The group's original building, a former merchant residence in what has become "Arts Square", still houses it today. In addition, the Sharjah Art Museum opened in 1997, again a project of the Sheikh, in a new building, but in the historical part of town, and designed with a traditional style (see Figure 7).

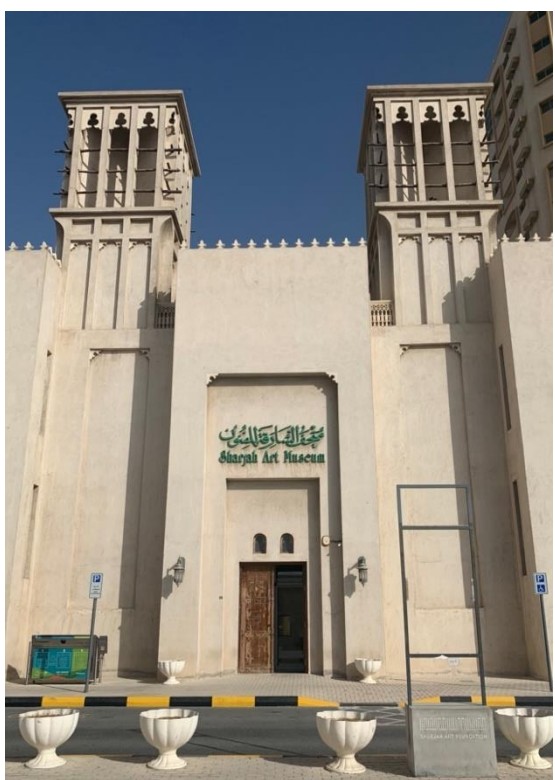

**Figure 7.** The Sharjah Art Museum. Photograph by the author.

That same year, the Sharjah Art Institute—dedicated to teaching—was established in the former Al Serkal House.[1] The Maraya Art Center—not linked directly to a specific royal family member but rather a project of the Qasba, a neighborhood and business center founded by the Sharjah Investment and Development Authority (or Shurooq)—was founded in 2006. It is a modern construction, but in the Mudejar style. The Sharjah Art Foundation opened its doors in 2009 in various buildings around Arts Square and Calligraphy Square. Originally, all these structures were in traditional edifices, but, in 2013, new construction was also completed in Al Mureijah Square to modernize these spaces all while keeping the historical aesthetic.

Since then, both the SAF and the Maraya Art Center, though in a lesser way, have expanded and added buildings to their rosters. Unlike Dubai and Abu Dhabi, where both have witnessed adding new, independent structures, or additional branches of foreign entities, Sharjah's art scene is exclusively homegrown and dominated by two institutions: SAF and Maraya. In 2015, the Maraya Art Center opened the architecture and design-specific 1971 Design Space at a second location (Flag Island). Like the first Maraya building, 1791 Design Space is modern; a well-known architect does not seem to have conceived it, or at least that information is not promoted, unlike the other local institutions. Before enumerating, in detail, the various venues and styles of the SAF, Sultan Sooud Al Qassemi, also a member of the royal family, created the Barjeel Art Foundation in 2010. This collection does not have a permanent physical space—a characteristic we observe more in Dubai (such as with the Atassi Foundation or the Dubai Collection) than in Sharjah—but is nearly always on view either at the Maraya Art Center, elsewhere in Al Qasba (the shopping district hosting the Maraya art center) or, since 2018, at the Sharjah Art Museum.

As for the Sharjah Art Foundation, one name means dozens of locations. At the time of writing, the SAF website breaks down its various buildings into two main categories: Adaptive Reuse and Constructed by SAF. Table 2 illustrates this information. Unlike Abu Dhabi and Dubai, most buildings employed by the Sharjah Art Foundation, which is arguably the largest art institution in the emirate, are historical—whether they be ancient fortresses, 19th-century merchant houses, or 20th-century industrial buildings (for example,

the Kalba Factory, Dibba Al Hisn Ice Factory, etc.). There is a strong emphasis on celebrating Sharjah's past from various points in its history. The "adaptive reuse" of historical residences (any building on the list that begins with "bait", meaning "house of" or "home") is a demonstration of this tendency. The way in which the SAF has reused or rebuilt other locations also manifests this trend.

**Table 2.** Table listing different construction types for the various locations of the SAF.[2]

| Construction Type | Location in Emirate | List of Buildings |
|---|---|---|
| Adaptive Reuse | Historical Neighborhood | Bait Al Hurma; Bait Abdulraheem Jasem; The Urban Garden; Bait Gholam Ibrahim Al Issa; Fen Restaurant; Fen Café; Bait Sultan Nasser Al Aboudi; Bait Hussain Makrani; Bait Habib Al Youssef; Bait Haiderabadi; Bait Al Ansari; Bait Khalid Bin Ibrahim Al Youssef; Information Centre; Dar Al Nadwa; Ceramic House; Bait Al Serkal; Bait Obaid Al Shamsi; Majlis Sheikh Mohammad and SAF Shop; Art Institute; SAF Main Offices; Collections Building |
| | Sharjah City | The Flying Saucer Sharjah; Old Radio Station; Memory Park; Sharjah Planetarium; Photography Gallery; Khalid Bin Mohammad School |
| | Al Hamriayh | Hamriyah Art Centre; Hamriyah Crown Bureau |
| | Kalba | Kalba Factory; Kalba Kindergarten |
| | Korfakkan | Khorfakkan Cinema |
| | Al Dhaid | Sheikh Khalid Palace Dhaid; Preventive Medicine Centre |
| | Al Madam | Sheikh Khalid Palace Madam; Madam Art Centre |
| | Dibba Al Hisn | Dibba Al Hisn Ice Factory |
| Constructed by SAF | Historical Neighborhood | Galleries 1–6 (each a separate construction in the Arts Area) |
| | Sharjah City | Rain Room Sharjah |
| | Al Hamriayh | Al Hamriyah Studios |

One of the most noteworthy examples of recycled buildings for the purposes of showcasing art is the "Flying Saucer." Dating from the 1970s, this unusual construction has hosted a variety of businesses such as a French-style *pâtisserie*, restaurant, gift boutique, news kiosk, and tobacco vendor—all in one, then as the Al Maya Lal's supermarket and Life Pharmacy, next as the home to the Sharjah co-operative society, and, lastly, as the Taza Chicken fast food restaurant (Sharjah Art Foundation n.d.). The SAF acquired the building in 2012, and since 2015 has used it as an off-site venue. A renovation project began in 2018, led by Mona El Mousfy of the locally based SpaceContinuum (*sic*) Design Studio (Yerebakan 2020). This firm renovated historical buildings in various galleries of the SAF around Arts Square, in addition to the modern building that houses the Rain Room (Middle East Architect 2018).

For the latter, the SAF has featured this immersive installation by Random International since 2018, but such institutions as the Barbican, London (2012); MoMA, New York (2013); Yuz Museum, Shanghai (2015) and LACMA, Los Angeles (2015–2017) (Selvin 2018) have also exhibited it. Indeed, Sharjah's art scene is host to a very local set of institutions, but its understated partnerships spread nearly as far and wide as those of Abu Dhabi. Among them are the Australia Council for the Arts, British Council, Danish Agency for Culture, Goethe Institut, Institut Français, Institut für Auslandsbeziehungen (ifa), Japan Foundation, Office for Contemporary Art Norway (OCA), and Wiener Festwochen. By collaborating in such partnerships, the Foundation receives the approval of this long list of international institutions, but the SAF does not publicize this information as much as Abu Dhabi does with the Louvre or the Guggenheim.

In addition to the building housing the Rain Room, another significant new structure is the Al Hamriyah Studios, located in the eponymous town on the northeast coast of Sharjah,

and inaugurated in 2017 (Universes n.d.). Local architect Khaled Al Najjar from the agency dxb.lab designed this modern structure on the site of an old souk. This fact highlights that even in the case of new constructions, Sharjah's architecture, especially when used for an artistic or cultural purpose, both reflects and celebrates the local past.

Beyond the art milieu, among the various buildings used to house the different entities of the Sharjah Museum Authority (or SMA), five are new, more modern-looking constructions built by local, unremarkable architects. Two are new constructions, built in the 1990s, yet made to look much older: the Sharjah Art Museum has been built in the old style to remain in architectural harmony with the other structures in the center of town. Likewise, Hisn Khor Fakkan is a new building but made to look like the former nearby eponymous fort. Like the SAF's use of pre-existing vernacular structures, the SMA uses four former 19th-century residences, an old fortress, a school, and an airport from the 1930s and, like the Al Hamriyah Studios, an old souk built in 1987.

## 5. Discussion: How Do the Three Emirates Interact and Compete Artistically?

The above section presents three different art ecosystems in proximity and how they have evolved differently. While Abu Dhabi remains known for its large, international institutions, Dubai for its alternative-chic galleries, and Sharjah for its homegrown Foundation, we observe architectural examples in each emirate that demonstrate an influence from the other two.

### 5.1. DUBAI—SHARJAH

As mentioned previously in the introduction, one unplanned legacy of Sheikha Fatima's establishment of the General Woman's Union in the 1970s is the later trend for other Sheikhas, following in her footsteps, to create non-profit art centers. The first one to officially open, Tashkeel, was founded by Sheikha Latifa bin Maktoum Al Maktoum (niece to the current ruler) in 2008 to continue making art after higher education and to provide studio space for local artists, as no such structure existed at the time. The building—still the home today—is nothing special. It is neither a refurbished warehouse nor a renovated residence. Like the Flying Saucer, it has gone through many versions, such as a supermarket, then a nursery, then part of the Latifa College for Girls. A year later, Sharjah Art Foundation opened its doors under the direction of another Sheikha, Hoor Al-Qasimi, again, the daughter of the current ruler—perhaps a coincidence of temporal proximity, perhaps a demonstration of how the juxtaposition of the emirates allowed for easy cultural exchange and inspiration.

### 5.2. DUBAI—ABU Dhabi

In Abu Dhabi, the Sheikha Salama bint Hamdan Al Nahyan Foundation (founded by the wife of the current ruler Sheikh Mohammad bin Zayed, son of Sheikh Zayed and Sheikh Fatima) opened its doors in 2010. Its missions span beyond Art to Culture and Heritage, Childhood Development, and Health. In 2015, however, the SHF followed in the footsteps of other initiatives in Dubai and Sharjah, by opening Warehouse 421. Perhaps the use of the word "warehouse" as part of its title is simply a coincidence, but let us remember that Alserkal Avenue, officialized in 2008 and opening its new expansion in 2015, had already received acclaim for rendering industrial warehouses more desirable in an untypically "Dubai bling" aesthetic. Alserkal Avenue followed neither the Sharjah model (for the most part, using former residences in the historical center) nor that of Abu Dhabi (creating outspoken modern buildings away from the traditional center). In the case of Warehouse 421, also breaking with Abu Dhabi tradition, it is composed of parts of renovated existing warehouses. It also has new constructions, but on torn-down foundations in Mina Zayed, an industrial port of the capital away from both Saadiyat Island and the Cultural Foundation (i.e., the historical town center). Lastly, the design was performed not by a large, international firm, but by the local GHD Consultants in Abu Dhabi. Since their respective opening dates, they have demonstrated a rivalry: copying

each other by providing similar offerings and support such as residencies, prizes, funding, exhibitions, etc. Competition, yes, but one that has helped push the local art scene forward.

*5.3. ABU DUBAI—DUBAI*

Conversely, while Dubai's art scene has mainly used vernacular structures (or in the case of the Alserkal expansion, new structures made to look like vernacular ones), the one exception at Alserkal Avenue is Concrete. Opened in 2017 and designed by the Office for Metropolitan Architecture (or OMA), founded by Pritzker-winning Rem Koolhaas. It is one of the first buildings specifically designed for displaying art and is still one of the few today. It is as if Alserkal Avenue, in positioning itself as a "mini Saadiyat Island" (as both are cultural districts), felt it necessary to have a "name brand" architect create a building there to compete, on a smaller scale, with Abu Dhabi.

Art Jameel is an exception to the fact that few buildings in Dubai have been built specifically for art. This museum is the local branch of a previously existing entity: Art Jameel, founded in 2003 by Community Jameel, the philanthropic organization of Saudi Arabian businessman Abdul Latif Jameel. While Jameel itself does not have the same international renown as the Louvre or the Guggenheim, before opening up a branch in Dubai, the group had already partnered with the Victoria and Albert Museum, The Prince's Foundation School of Traditional Arts (established by the current King Charles III, then the Prince of Wales), the Delfina Foundation, and the Metropolitan Museum of Art. Art Jameel Dubai opened its doors in 2018 in the Jaddaf Waterfront area of the city. Unlike the institutions in Abu Dhabi or Sharjah, which were the initiatives of local leaders and government entities, in this case, the Saudi Arabian Jameel family approached the Dubai government about opening a branch of Art Jameel in that city. For the family, Dubai was an obvious choice as there was a growing art market there, yet no large art museum (Carver 2020). It is no wonder that the Government of Dubai agreed to allow the family to build another museum there: as competition between emirates is non-stop, it was imperative to keep up with its neighbors.

The family's choice of architect, Serie Architect, shows the established relationship the firm has had with the Jameel family: Serie had previously proposed to build the Bayt Al Fann as part of Jameel Art in Jeddah in 2014. The family did not want a distinguished "outspoken" structure, but rather something understated, perhaps purposefully in contrast with the larger institutions in Abu Dhabi (Carver 2020).

Likewise, the Etihad Museum was not intended to display art, but it was constructed as a museum. Designed by Moriyama & Teshima Architects (or MTA), this structure was first conceived as a general history museum showcasing the heritage of the UAE—"Etihad" itself means "union or alliance" and refers to the various emirates coming together as the UAE in 1971. It is also adjacent to the "Union House", where the country's founding fathers signed the document establishing the federation. The Government of Dubai's selection of these architects, unlike Abu Dhabi's choice of starchitects or Sharjah's choice of more local firms, demonstrates Dubai's endeavors to differentiate itself from the other two states. Furthermore, MTA was also responsible for the Aga Khan Museum (in Toronto), which has had a "close relationship with the UAE since 2014" (Aga Khan Museum 2017). The Friends of the Aga Khan Museum launch in the Gulf and South Asia took place at Concrete at Alserkal Avenue, Dubai—not in Abu Dhabi, the capital—in 2017 (Shaikh 2018). Concrete was also shortlisted for the Aga Khan Award for Architecture that year, and the MTA group also designed the Canada Pavilion for the Expo 2020 and participated in the competition for the Dubai Archaeology Museum. Perhaps for the Government of Dubai, it was a priority to employ architects whose design was already accepted and approved for an institution showcasing Islamic art and culture elsewhere; this, then, could have motivated them to do so for the first museum constructed in its history for that purpose.

The building opened in 2017 as a space to feature exhibitions on the history and culture of the country. In 2021, Dubai Culture announced that the Dubai Collection—"the first institutional collection of modern and contemporary art in Dubai . . . launched by Dubai

Culture & Arts Authority DCAA, in partnership with Art Dubai Group"—would have its first exhibition When Images Speak, at the Etihad Museum (Dubai Collection 2021).

In both cases, with the Jameel Art Centre and the Etihad Museum as a host for the Dubai Collection, the Government of Dubai had a considerably less active role than that of Abu Dhabi or Sharjah with their respective museums. Again, for Jameel, the family (who already had an arts center elsewhere that had international partnerships and, thus, international approval) approached the government of Dubai about the project, whereas for the Dubai Collection at the Etihad Museum, first the building was made and then the collection (a hybrid collection of works on loan from both private individuals and corporations) was exhibited there. This is not to say Dubai does not value the arts, but to highlight how, for most of its history, the key players developing the art scene were individuals rather than government initiatives or royal family members, unlike in the other two emirates.

### 5.4. ABU DUBAI—SHARJAH

Eventually, even for Sharjah, the appeal of world-renowned architects could not be resisted. In 2018, Shurooq announced that a new structure would be built to physically commemorate the emirate's designation as the 2019 UNESCO Book Capital. Donned the nickname the "House of Wisdom", this edifice's purpose is more in the literary realm than that of the visual arts, its inclusion in this article is important given its similarities to both earlier institutions in Sharjah, as well as the choice of architect.

Like the Sharjah Art Foundation's strong grasp on the art scene through the hands of Sheikha Hoor Al-Qasimi, literary activities in that emirate are under the leadership of her sister, Sheikha Bodour Al-Qasimi. Sheikha Bodour founded the local publishing house the Kalimat Group, along with its eponymous philanthropic foundation. Additionally, she established the Emirates Publishers Association, the UAE Board on Books for Young People, and serves as the President of the International Publishers Association. She is also the Chairperson of Shurooq. In Sharjah, the idea of keeping cultural power in the family is of utmost importance. The Sharjah Investment Group commissioned none other than Norman Foster for the "library of the future", (Waite 2021). Two facts are noteworthy here: first, that this structure is a commission and not the fruit of an architectural competition demonstrates the city's desire to have that specific architect who had already designed the yet-to-be-built Zayed National Museum in Abu Dhabi. Foster + Partners, in addition to structures in the cultural realm, have also designed several other buildings in Dubai such as the Index Tower, certain parts of d3, the Apple Store at the Dubai Mall as well as the Central Market/World Trade Centre Souk, Masdar City and Institute, all in Abu Dhabi, as well as the UAE Pavilion at the Milan Expo in 2015. In 2016, the group also designed the Investment Corporation of Dubai (or IDC) Brookfield Place, near the DIFC, which, since 2021, is home to the Dubai branch of the illustrious London-founded Arts Club. Thus, while the Zayed Museum for which Foster + Partners is perhaps more known in the UAE has yet to be built, the firm has a long reputation there, notably in Dubai and Abu Dhabi. It seems that the choice of this firm for the new House of Wisdom—their "library of the future"—was a demonstration of a new era in Sharjah's cultural timeline, one that demonstrates greater influence from outside emirates, and looking to the future and not the past.

Indeed, her younger sister Sheikha Hoor (of the SAF) followed suit by commissioning Adjaye Associates to design the new Africa Institute, over which she presides. The Institute's focus—Africa and African diaspora studies—falls beyond the realm of contemporary art, like the House of Wisdom. However, in the interest of highlighting a seemingly growing tendency in Sharjah to employ well-established architects, we have included it in our study. The design was first revealed in 2021 and the Institute is set to open in 2023. Once again, this is not the first work the Ghanaian-British architect has designed for the UAE: in 2019, the Abu Dhabi government commissioned him to design a tripartite multifaith complex—Abrahamic Family House—on Saadiyat Island. This fact again suggests it was

important for Sharjah to employ the same architects who had already built other art-focused structures in Dubai and Abu Dhabi in order to architecturally compete with them.

### 5.5. SHARJAH—DUBAI/ABU DUBAI

While Dubai and Sharjah, as we have seen, began to have purpose-built structures by brand-name architects, conversely, Abu Dhabi and Dubai followed in Sharjah's path as each, with time, began to use more historical buildings for museums and other institutions to highlight their past.

In Dubai, for example, the Al Shindagha Museum first opened as a museum in 1997 without any renovation and closed in 2015 for massive reconstruction. The firm Shankland Cox—also responsible for the Alserkal Avenue expansion—won the winning design, and the new renovation has begun to reopen in small phases since 2019. We can view this structure housing multiple museums in the historical part of Dubai as that emirate's version of Sharjah's "Arts Square", where many of the historical buildings are part of the SAF or the Sharjah Museum network but have been highly renovated. In addition, the Saruq Al Hadid, housed in the house of the former Sheikh Juma bin Maktoum al Maktoum, also in the Shindagha Heritage District, showcases the archaeological discoveries of the Saruq Al-Hadid site. This facility opened in 2016.

As for Abu Dhabi, the Al Qattara Art Space (a renovated version of the Bin Ati Al Darmaki residence) opened in 2011 as an arts education center and gallery. Furthermore, the renovated Qasr Al Muwaiji and aforementioned Qasr Al Hosn (both former royal palaces) opened as museums in 2015 and 2018, respectively. In the case of the Qasr al Hosn, the renovation was given to Danish and Abu Dhabi-based firm CEBRA Architecture, breaking with the local tradition of using more "name-brand" architects.

It is not surprising that Abu Dhabi and Dubai present fewer tangible examples of following in Sharjah's footsteps because both have a distinct image that does not match Sharjah's ideals. Sharjah is the smaller, less populated, and poorer of the three. It is more traditional and far more conservative, and the ruling family issues from a different tribe (the Al Qassemi) than those of Abu Dhabi and Dubai (both of whose royal families, the Al Nahyan and the Al Maktoum, respectively, belong to the Bani Yas Tribe).

However, architecture aside, Sharjah's influence on the other two emirates can be seen not less through its buildings than in intangible ways that demonstrate an increasing support for and value of the arts and culture. While all three states have had an art scene since the country's early days, Sharjah was the first to have strong government support for the arts and specific government entities to celebrate the arts in all forms, culture, and heritage. Indeed, in this small emirate, active government participation and support have been taking place since the 1980s, and its larger neighbors only officialized their respective culture bureaus (which were also at times fused with the tourism offices) in the early 2000s.

### 5.6. DUBAI—ABU DUBAI

Finally, there is a similar influence that Dubai has had on Abu Dhabi's art scene in that, with time, there are more buildings used for small, grassroots galleries and foundations in residences, hotels, and other towers. Table 3 lists various galleries or non-profits based in Abu Dhabi that are located in hotels, villas and other kinds of structures not originally indented for artistic use. The architecture itself is not what is noteworthy, but rather the use of the space as a new concept for that city. Below is a list of galleries and foundations that did nothing exceptional by choosing to open a gallery in a space designed for a far different purpose. What is significant is that in Abu Dhabi, it is relatively recent compared to when this happened in Dubai.

This tendency suggests Dubai's influence on the Abu Dhabi art scene, not architecturally, but socially by using vernacular examples of otherwise "non-artistic" structures.

**Table 3.** Abu Dhabi-based galleries located in hotels, villas, towers, or office space.

| Name of Gallery or Foundation | Name or Type of Building | Date of Establishment |
| --- | --- | --- |
| Salwa Zeidan Gallery | Millennium Capital Gate Hotel | Opened 2004, reopened 2009–Present |
| Qibab Gallery | Al Bateen Area, Villa 3, Street 15 | c. 2008–2011 |
| Al Arjun Gallery | Mixed Office Space in Al Tibbiya | 2008–Present |
| Contempo Corporate Art | Emirates Palace Hotel | 2008–2014 |
| Acento Gallery | Warehouse, Al Meena Port Area | 2010–2011 |
| Ghaf Gallery | Villa (Al Kaleej Al Arabi Street) | 2010–2014 |
| Barakat Gallery | Emirates Palace Hotel | 2010–2016 |
| UR Gallery | Nation Towers | 2014–Present |
| N2N Gallery | Al Ain Tower | 2013–Present |
| Etihad Modern Art Gallery | Al Bateen Area Villa 15, Al Huwelat Street | 2013–Present |
| Bait 15 | Villa (Al Salam Street) | 2017–2021 |
| Engage 101 | Hosted shows at Bait 15/Online | 2020–Present |

## 6. Putting Art, Architecture, and Society of the UAE in a Larger Context

With some exceptions, analyzing the architecture of the art scene in the United Arab Emirates reveals societal facts and demonstrates the multiple waves of influence that have defined its architectural artistic landscape today. Through various buildings in Abu Dhabi, Dubai, and Sharjah, we witness a largely nomadic past with some historical instances demonstrating a military or commercial precedent (ancient forts or merchant villas), the presence of industry (warehouses and factories—especially in Dubai and Sharjah, who have always had less oil than Abu Dhabi), buildings designed to welcome the influx of a foreign labor force and the rise, of Dubai especially, as a business and finance hub (malls, skyscrapers, and hotels), and, finally, purpose-built structures to house the art scene that began in the aforementioned types of building. These later examples, purpose-built museums such as the Louvre Abu Dhabi or exhibition spaces such as Concrete at Alserkal Avenue, are the examples that more recently have put the country on the international art world radar and are examples of academic architecture. In contrast, the other examples are, in an Emirati way, vernacular. Emirati artistic vernacular, perhaps.

Historically the term vernacular has been applied to more "primitive" examples of architecture designed without architects, such as prehistoric theaters, ancient houses far older than those cited in this article, age-old burial grounds (cemeteries, necropolis, etc.), or troglodyte structures (Rudofsky 1964), among others, associated with folk or tribal culture. One can also apply it, however, to examples of architecture in the UAE that have "everyday" use, such as 20th-century villas, industrial buildings, hotels, malls, and towers (Christenson 2011; Upton and Vlach 1986). Given this, some of the more historical merchant houses or forts used, while chronologically later than more widely accepted examples of vernacular architecture, are examples of traditional architecture as they are some of the oldest in the country, given that the region has a nomadic past. In short, these are unrecognized expressions of vernacular architecture as they stray from the traditional definition of the term.

For the vehicular or academic examples of architecture, these purpose-built structures have been, for the most part, commissioned by members of the royal families of each emirate or government committees or offices run by such individuals (the Cultural Foundation, anything on Saadiyat Island, the Etihad Museum, later construction for the SAF) or wealthy families (Alserkal Avenue's modern buildings and Concrete, the Jameel Art Centre). There is an association between vehicular architecture and local power structures (political, tribal, financial, etc.).

By most definitions, vehicular and vernacular, or as also referred to, organic and technocratic (Bacon 2005) oppose each other and compete, yet in the case of the United Arab Emirates, the two sometimes overlap and even perpetuate the other. This can be seen in such examples as the expansion at Alserkal Avenue, Warehouse 421, or various examples of the Sharjah Art Foundation are examples of this concept: these are recent structures designed to look historical, or historical and industrial buildings that have kept their aesthetic but have been renovated internally to modern standards.

Another way to look at different examples of the vernacular, and by comparison vehicular, architecture in the UAE is by analyzing these buildings through Edward Soja's Thirdspace theory:

> . . . everything comes together . . . subjectivity and objectivity, the abstract and the concrete, the real and the imagined, the knowable and the unimaginable, the repetitive and the differential, structure and agency, mind and body, consciousness and the unconscious, the disciplined and the transdisciplinary, everyday life and unending history. . . . I define Thirdspace as an-Other way of understanding and acting to change the spatiality of human life, a distinct mode of critical spatial awareness that is appropriate to the new scope and significance being brought about in the rebalanced trialectics of spatiality–historicality–sociality. (Soja 1996)

It is hard to put various instances of architecture used for art in the UAE in just one category. They can be at once vernacular and vehicular, historical or modern, tangible or intangible (for all the buildings announced yet never built). Some examples, such as the Louvre Abu Dhabi, are without precedent and revolutionary, while others repeat past trends such as the 19th-century residential aesthetic, or warehouses.

## 7. Conclusions and Future Study

This article has presented examples of architecture used in the Emirati art scene, whether these structures were purpose-built or adapted preexisting structures, and what they reveal about each state's art scene and, later, the relationship between Abu Dhabi, Dubai, and Sharjah. I have argued that due to persistent competition between these emirates, each one has had an initial version of an art scene and that with time each state has influenced the other two due to this rivalry. Abu Dhabi's art scene began in large, state-run and starchitect-designed institutions, while Dubai's more grassroots artistic beginnings were first housed in people's homes (or other residential structures) and later on in industrial hangers, whereas Sharjah has tended to renovate or reuse preexisting historical structures for its various public foundations and museums. As each scene has evolved, we observe a borrowing or appropriation of architectural elements found originally in the other states. Given the increasingly blurred lines between the different states' kinds of architecture, following Soja's theory of the Thirdspace, it is helpful to not think of the local architecture as vernacular or vehicular, but eventually a fusion of both.

Further research on this subject could include a more detailed analysis of different award-winning architects' work in each emirate beyond the art scene, or delve even further, beyond the UAE—whether the work has been completed or not. Such examples include Jean Nouvel's W Hotel in Dubai, the Dubai Opera, the National Museum in Doha, the Doha high-rise office tower and his hotel in Al Ula; Zaha Hadid's Opus Hotel in Dubai, that city's Financial Market, the Bee'ah headquarters in Sharjah or the Sheikh Zayed Bridge in Abu Dhabi; or lesser-known architects such as CEBRA's contributions to various buildings in the three emirates, or that of the Dubai-based Killa Design. Moreover, an in-depth study could monitor the buildings or collaborations that were announced but never realized. In addition to those on Saadiyat Island, these include the British Museum's participation in some exhibitions at the DIFC in the early 2000s or the cultural district that Dubai never built along the creek that would have housed the Museum of Middle Eastern Modern Art (MOMEMA).

As the United Arab Emirates is just barely fifty years old, its art scene, architectural landscape, and local society are still in their nascent days. With both the country's rapid

growth and constant change since its early days, while ambitious, it is important to track the evolution of its art, architecture, and society, together from the beginning to reveal important facts about their origins and relationship.

**Funding:** This research received no specific external funding, but the author has benefitted from funding for past research trips by the Institut d'histoire moderne et contemporaine (IHMC), UMR 8066; Artl@s (Ecole normale supérieure—PSL); and the Bourses Mobilité Île-de-France.

**Data Availability Statement:** Not applicable.

**Acknowledgments:** The author wishes to thank her thesis director Joyeux Prunel for her ongoing guidance for this project, as well as the teams at the Institut d'histoire moderne et contemporaine and the Ecole Doctorale 540 of the Ecole normale supérieure—PSL, for their support.

**Conflicts of Interest:** The author declares no conflict of interest.

## Notes

[1]    No relationship to the Dubai-based Al Serkal family, their foundation, or Alserkal Avenue.

[2]    See the Sharjah Art Foundation's website on their "Venues, Sites, and Architecture", for more information, here: https://sharjahart.org/sharjah-art-foundation/about/sharjah-art-foundation-venues, accessed on 10 January 2023.

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
