# Peer review of "The Emirati Vernacular: Tracing the UAE’s Art History through Architecture as a Reflection of Society"

_arts, 2023_

Round 1

Reviewer 1 Report

This article provides an excellent overview of the art scene in a particular geographic area. It is contemporary and would be a useful resource for potential artists wanting to exhibit in the area. 

What could be strengthened is some details and more of a critique: a clarity of size/scale of spaces, whether these are public (free entry) or public or commercial; presence of artist-run spaces; the success of foreign architect designed spaces; sustainability. 

The 'society' component could be strengthened also: whether these architectures create a vibrant public events scene, level of imposition of curatorial guidelines etc.  

Author Response

Dear Reviewer 1,

Thank you for taking the time to read and comment on this article. I have gone through and made many of the suggested changes. I am wondering with this new version if you still deem it necessary to add the following information:

  1. A clarity of size/scale of spaces: I have added km2 for various locations. Do you think it is necessary to do so for each individual building discussed?)
  2. Whether these are public (free entry) or public or commercial: this was actually included in the first table, do you think it would be better to state this more clearly when discussing each emirate and each building? I fear that that could get quite lengthy. 
  3. Presence of artist-run spaces: there are no artist-run spaces discussed in this article and, as far as I know, the ones that did exist no longer do. Should that be included? 
  4. The success of foreign architect designed spaces: how would one measure this? And what do we define as foreign? Most people living in the UAE are not from there.
  5. Sustainability: I agree this could be an interesting addition, but again, would that add on too much to the article?
  6. Whether these architectures create a vibrant public events scene, level of imposition of curatorial guidelines etc.: I agree this could be an interesting addition, but again, would that add on too much to the article if I do so for each building?

Thank you again for your guidance and advice!

Kind Regards,

Eve

Reviewer 2 Report

This is an insightful article from a historical point of view. Authors may want to use an analytical method to stress the relationship between art and the  architectural relationship between art, society and the architecture. 

Author Response

Dear Reviewer 2,

Thank you for taking the time to read and comment on this article. I have gone through and made many of the suggested changes. Please let me know if you have any further comments or questions. 

Thank you again for your guidance and advice!

Kind Regards,

Eve

Reviewer 3 Report

Peer Review Report for Authors

First: Manuscript Information

Article ID

arts-2222889-peer-review-v2

Manuscript type

Article

Journal title

Arts

Title of the manuscript

The Emirati Vernacular: tracing the history of art in the UAE through its architecture as a reflection of society

Second: General Comments

1.    Main Impressions of the Article

§  I want to share my general good impressions with the authors from the beginning.

§  The research title intrigued me because of its benign environmental, social, economic, and human dimensions, which societies require to sustain and improve their quality of life.

§  The history of art in the UAE through its architecture is an excellent approach to studying society. It provides a good acceptance as a research line at multiple levels, highlighting its significance.

§  Therefore, after getting a good first impression, I was drawn to review this study.

Therefore, I thank the authors for this effort; and set the following comments to enhance the end shape of this required research.

2.    Statistics

            Pages: 21                     References: 24             Paragraph: 243

3.    The Uniqueness of the Paper and its Right to Research

§  The topic (tracing art history through architecture) is not unique.

§  The keywords (Emirati art scene; manifestations of power through art and architecture; vehicular vs. vernacular space; grassroots art scenes; women in the arts; royal families in the arts; arts in the Gulf) are frequently addressed in the current state of the art.

§  Given the importance and continuity of the research point, it is worth investigating.

§  Therefore, the research gate is open to this research point regarding all surrounding conditions in this field.

4.    Article Summary

This article examines the evolution of the art scene in the three largest emirates of the United Arab Emirates from the foundation of the federation through today as a reflection of local societal truths. Each emirate has developed an art scene unique from the others, and each has been housed in different kinds of mostly vernacular architecture. These structures provide a rich discussion on what is considered vernacular in a modern context and where the definition of one stop and the other begins.

5.    Completeness of the Article Structure (Presence/Absence of Main Sections)

Research articles generally should follow a standard structure in which information is presented. This allows readers to quickly find the information needed by looking it up in the relevant section. However, the structure varies for articles, style guides, journal requirements, etc. So, ensuring that a manuscript is organized logically and consistently with all required sections is critical. According to the instruction of this journal, the structure should include an Abstract, Keywords, Introduction, Materials and Methods, Results, Discussion, and Conclusions (optional) sections, with a suggested minimum word count of 4000 words. So, please refer to the journal webpages for specific instructions and templates, which are available at:

https://www.mdpi.com/journal/arts/instructions

Therefore, please carefully check the following comments for this manuscript to ensure the completeness of the structure:

§  The following sections are presented in the manuscript:

É  Title.

É  Abstract.

É  Keywords.

É  Introduction.

É  Discussion.

É  Conclusion.

É  References.

§  The following section was absent from the manuscript:

É  Literature Review: It is preferred to separate it.

É  Materials and Methods.

É  Results.

More organization would be helpful to align this paper with the journal's guidelines.

6.    The Robustness of Theoretical Analysis in this Article

§  Based on the proposed motivation for the research and formulation of research questions, and addressing the various parameters and research topics, it appears that it fits the nature of the problem addressed and the investigated question.

§  The theoretical analysis in this article falls short of my expectations for robustness, with only around 65% of its arguments meeting the required standard.

§  This item is acceptable and can be passed.

Third: Advantages& Disadvantages of the Articles' main sections

1.    Article Title

§  It came in 19 words.

§  It may be shortened.

§  Authors may look at the proposed following title:

"Tracing the UAE's art history through architecture as a societal reflection".

2.    Abstract

The abstract consisted of 180 words and was appropriate but fell short of conveying complete information about the research. As an incomplete overview, it failed to provide sufficient details and did not address the research question as expected for this type of article.

However, the authors need to reconsider and show the following key sections, which are required in a typical abstract, they are:

(a)   Background or Introduction: This section should provide a brief context and explanation of the problem or research question being addressed in the study.

(b)   Objective(s): This section should clearly state the study's main objective(s), including any hypotheses or research questions being tested.

(c)   Methods: This section should describe the methods or approach used to conduct the study, including the study design, data collection, analysis, and any statistical techniques used.

(d)   Results: This section should summarize the key findings or outcomes of the study, including any statistical analyses or significant results.

(e)   Conclusion(s): This section should summarize the study's main conclusions or implications, including any limitations or future research directions.

3.    Keywords

There are ten suitable and accepted keywords.

4.    Introduction

The authors provided an uninformative introduction, which did not clearly outline the research's context and motivation. Additionally, the phrasing of the research questions may lack the traditional format that is commonly recognized. The introduction did not conclude with a clear and accurate disclosure of the article's objective.

The typical structure of this section (introduction) typically could be broken down into the following components:

§  Background information: This provides context and sets the scene for discussing the topic. It includes information about the problem or issue that the article will address, the scope of the problem, and any relevant historical or social background.

§  Literature review: This summarizes the existing research on the topic and identifies gaps in knowledge or areas where further research is needed. It should include references to other relevant studies and provide a framework for the article.

§  Objectives and research questions: This outlines the study's specific aims.

§  Methodology: The methodology section should provide a clear and detailed description of the research process, allowing other researchers to replicate the study if needed. It should also demonstrate the validity and reliability of the research findings.

§  Structure of the article: This provides an overview of the article's structure, including the sections that will follow and what each section will cover.

Overall, the introduction section of an article should provide a clear and concise overview of the topic and the research presented in the article. It should engage the reader and make them interested in reading further.

5.     Literature Review

There is no independent section for reviewing the literature. It is an important section that is preferred to be present in the research article, as it relates to the following objectives:

It appears that the author(s) did not address the historical sequence from the broader context to the exact point of the investigation, which did not give the reader a logical sequence. So, authors should increase the literature review, and it is recommended to look at the suggested references listed in the references section.

The structure of a literature review section can vary depending on the specific requirements of the article, but here are some typical components:

§  Introduction: The introduction should provide an overview of the topic and its importance. It should also introduce the purpose of the literature review and the research question(s) that the review seeks to answer.

§  Search strategy: The search strategy should detail how the literature review was conducted, including the databases, search terms used, and any inclusion and exclusion criteria applied.

§  Synthesis: The synthesis should summarize the key findings from the literature and highlight the main themes and trends. It should also identify gaps or inconsistencies in the literature and explain how the review addresses them.

§  Critique: The critique should evaluate the quality and relevance of the literature, highlighting its strengths and weaknesses. It should also consider the methodological limitations of the studies included.

§  Conclusion: The conclusion should summarize the review's main findings and provide a clear answer.

§  References: The references should include a complete list of all the sources cited in the literature review, following the appropriate citation style guidelines.

6.    Materials and Methods

Despite being a key section in article writing, this article did not contain a separate "Materials and Methods" section.

However, the "Materials and Methods" section of an article typically includes the following components:

§  Study Design: A brief description of the overall study design, including the type of study (e.g., observational, randomized controlled trial), the population or sample size, and any inclusion/exclusion criteria.

§  Participants or Subjects: A description of the participants or subjects included in the study, including relevant demographic information such as age, gender, and any medical conditions.

§  Data Collection: A description of the methods used to collect data, including any instruments or tools used for data collection (e.g., surveys, questionnaires, physical measurements), the duration of the study, and any procedures followed.

§  Variables and Measurements: A description of the variables being measured, including the primary outcome(s) and any secondary outcomes, as well as the methods used to measure these variables.

§  Data Analysis: A description of the statistical methods used to analyze the data, including any software or tools used and any assumptions made about the data.

§  Ethical Considerations: A statement regarding ethical considerations for the study, including any institutional review board (IRB) or ethics committee approval, informed consent procedures, and any measures taken to protect the privacy and confidentiality of study participants.

§  Limitations: Discuss study limitations, such as sample size, selection bias, or measurement error.

§  Reproducibility: A description of how the study can be reproduced, including any instructions on accessing the raw data or materials used.

§  Overall, the "Materials and Methods" section should provide a clear and detailed account of how the study was conducted so that others can understand and potentially replicate the study's findings.

7.    Results

The paper did not contain a separate section for results. It is needed to cover the proposed scientific addressed problem.

The "Results" section is essential to any research article presenting the study's findings or experiment. This section aims to report the results clearly, concisely, and organized, allowing readers to understand and interpret the data. Here are the typical components of a "Results" section:

§  Overview statement: Start with a brief introduction to provide context for the results. This should summarize the main findings and give a general idea of the outcomes.

§  Data presentation: Present the data in an organized and logical manner, often using tables, figures, and charts to effectively display the results. Ensure each visual element is properly labeled and has a clear title and caption to explain its purpose.

§  Narrative description: Accompany the visual elements with a written description of the results. Explain how the data was analyzed, and highlight the key findings and trends. Use clear and concise language to describe the results without interpreting or explaining them.

§  Statistical analysis: Provide details of the statistical methods used to analyze the data, including any tests performed, significance levels, and confidence intervals. Report the results of these analyses, ensuring that the statistical terms and notation are used correctly and consistently.

§  Subsections: If your study has multiple parts or components, use subsections to break up the results section into smaller, more manageable pieces. This will make it easier for readers to follow and understand the main findings.

§  Referencing figures and tables: Always refer to the figures and tables in the text of the results section. This will help guide the reader and ensure they can easily locate and understand the visual elements in context.

Please, remember that the "Results" section should only report the findings of your study without interpreting or discussing them. The interpretation and discussion of the results should be reserved for the "Discussion" or "Conclusion" section of your article.

8.    Discussion

This article section contains details and an analysis of the proposed study. It offered how the three Emirates interact and compete artistically. It addressed three different art ecosystems in proximity and how did they evolve differently?  It is detailed in six axes to observe architectural examples in three different art ecosystems in proximity.

§  These six axes presented an insightful discussion highlighting the study's question addressing the research issues and limitations, and it ended the study with a simple view of future cities.

§  However, it is preferred to cover studies with similar or different results.

9.    Putting Art, Architecture, and Society of the UAE in a larger Context

§  This section of the study is informative and outlines its points.

10.           Conclusions and Future Study

This section summarized the previous sections, answered the study's objectives, described its contribution, detailed its limitations, and provided recommendations for future research.

An article's "Conclusion" section plays a crucial role in the writing process by summarizing the key points, presenting valuable insights, and making a lasting impact on the reader. Typically, a well-organized conclusion encompasses the following elements:

§  Restatement of the main argument: Begin by briefly restating your article's main argument. This helps to remind the reader of the focus and purpose of your work.

§  In conclusion, the evidence presented in this article highlights the significance of the main argument.

§  Summary of key points: Provide a succinct summary of your article's key points or findings. This allows the reader to quickly recall the most important aspects of your work.

§  Throughout the article, we discussed [key point 1], [key point 2], and [key point 3], which all contribute to our understanding of [main argument].

§  Implications: Explain the implications of your findings or arguments and how they relate to the broader scope of the topic. This can include potential consequences, applications, or future developments.

§  The results of this study have important implications for [broader context] as they shed light on [specific implications or consequences].

§  Limitations: Acknowledge any limitations or shortcomings of your work. This shows that you are aware of the potential weaknesses of your arguments and helps to establish credibility.

§  While the findings of this article are compelling, it is important to note that [limitations or shortcomings], which may affect the generalizability of the results.

§  Recommendations or future research: Suggest further research or action based on your conclusions. This can help to inspire new ideas and guide future work in the field.

§  In light of these findings, future research should explore [recommendations or areas of future research] to further our understanding of [main argument].

§  Closing statement: End with a strong closing statement that leaves a lasting impression on the reader, often by connecting your work to a larger context or emphasizing the significance of your findings.

§  Ultimately, this article demonstrates the crucial role of [main argument] in [broader context or issue], underscoring the need for continued research and attention.

§  Remember to adapt these components to your article's specific requirements and style, ensuring that your conclusion effectively wraps up your work and leaves a lasting impact on the reader.

§  Consequently, the conclusion and recommendations of this article require to be revised.

Fourth: Improving the Article

1.    Language Quality

Language quality includes many issues in grammar, mechanics and style, readability, and vocabulary issues, as follows:

§  Grammar: 162 issues.

§  Mechanics and style: 116 issues.

§  Readability: 147 issues.

§  Vocabulary: 22 issues.

As a result, there are several issues with the language that needs to be revised.

2.    References

Proper referencing of other works used to build on the study is important; it is recommended that sources cited be more recently published. However, as the pace of scientific development differs across fields, each source must be evaluated on a case-by-case basis. Unless citing seminal research that may be quite old but still relevant, it is significant to ensure that recent sources (less than 10 years old by publication date) are cited when available. We have categories references as below:

Recent references: References refer to sources that are less than 10 years old by publication date.

Outdated references: References refer to sources over 10 years old by publication date.

Unknown References: References refer to sources that do not have a year of publication.

§  However, references are slightly small in number, and they can be accepted in historical aspects (distribution by year).

§  This study may need more references to be embedded and used.

§  The authors used a suitable reference management tool.

As indicated above, the authors recommended reviewing the following references to boost some research sections.

Derderian, E. Engendering Change: Charting a History of the Emirates through Women Artists. Hawwa. 2021, 19, 28-50, doi: https://doi.org/10.1163/15692086-BJA10016.

Konbr, U.; Abdin, A. Conservation as an Approach to the Sustainability of the Architectural Heritage: A Proposed Methodological Framework for Conservation. In Proceedings of the Second International ‎‎Conference of Architecture and Urban ‎‎Planning Departments, ARUP 2008, Cairo, Egypt, 25-27 October 2008; pp. 1-20.

 El Amrousi, M.; Elhakeem, M.; Caratelli, P. Reinterpreting Tradition; Hybridization and Vernacular Expression in Emirati Housing. IOP Conference Series: Materials Science and Engineering. 2019, 603, 052045, doi: https://doi.org/10.1088/1757-899X/603/5/052045.

Rashid, M.; Ara, D.R. Between Anticipative and Iconic: Re-imaging the Emirati Villa and its Spatial Assemblages. Journal of Planning History. 2022, 22, 47-67, doi: https://doi.org/10.1177/15385132211061816.

Benslimane, N.; Biara, R.W. The urban sustainable structure of the vernacular city and its modern transformation: A case study of the popular architecture in the saharian Region. Energy Procedia. 2019, 157, 1241-1252, doi: https://doi.org/10.1016/j.egypro.2018.11.290.

3.    Tables

Tables help authors concisely present detailed results, complex relationships, patterns, and trends. Every visual element included in the manuscript should be mentioned in the text. There are chances that tables may not be printed if they have not been cited in the text. This check helps authors identify whether all tables included in the manuscript have been cited.

§  In the manuscript, there are 3 tables.

§  Tables 1 and 2 are correctly cited.

§  Table 3 needs a cross-reference.

4.    Figures

Figures help authors concisely present detailed results, complex relationships, patterns, and trends. Every visual element included in the manuscript should be mentioned in the text. There are chances that the figures may only be printed if they have been cited in the text. This check helps the authors identify whether all figures included in the manuscript have been cited.

§  In the manuscript, there are 7 figures.

§  Figures are not correctly cited.

§  Citing these figures did not appear in the manuscript.

§  In this regard, all figures must be mentioned in the manuscript's text.

5.    Inclusive Language

§  The manuscript was checked for racist, sexist, and abusive language.

§  The manuscript does not contain language considered offensive or non-inclusive in publication.

§  In this regard, it is passed.

6.    Manuscript Wording

§  Using WORD in editing is good regarding fonts, paragraphs, formatting, etc.

§  The manuscript wording can be improved to reach the advanced level of this prestigious journal.

§  The Headings should be corrected to be hierarchical, Heading 1, Heading 2, etc., so they can appear in the Navigation Pane, helping readers easily navigate the article.

7.    Obeying the Journal Template

"Instructions for Authors, Submission Checklist, and the Microsoft Word template" are to be obeyed. Their link is:

https://www.mdpi.com/journal/arts/instructions

8.    Declaration of Conflicts of Interest

Declarations help the editorial board assess any real or perceived conflicts of interest and biases, facilitating fair manuscript evaluation. If there is no disclosure, this should also be mentioned. Check the submission guidelines for the accuracy of the format and placement of this declaration.

§  There is no statement regarding financial associations or competing interests in this manuscript.

§  The author did not disclose that there are no Conflicts of Interest or financial or personal relationships with people or organizations in their manuscript, which inappropriately influences his work and hinders transparency.

9.    Data Access Statements

§  No data access statements have been identified in your document.

§  Please ignore if your research paper does not involve or need data statements.

10.     Ethical Use of Images

§  The manuscript does not contain images of clear human participants.

§  This point is passed.

11.     Ethical Declaration

Ethical declarations are an integral part of the manuscript submission process. If the study involves human subjects, the author should declare whether written informed consent was obtained. Similarly, relevant details should be provided if the manuscript includes case reports/case series.

§  In this manuscript, no ethical declarations are included.

§  It may be that the article does not require an ethical declaration.

However, please verify the submission guidelines on these declarations' format, inclusions, and placement.

12.     Corresponding Author Details

§  Referring to "Instructions for Authors" correspondence details seem it be not set in this manuscript.

§  Additionally, please check the submission guidelines for specific requirements, such as blinding.

13.     Self-Citations

§  Citing one's own work is inevitable, especially when building upon previous research in a new article.

§  However, including many self-citations in a manuscript is discouraged.

§  This practice may represent excessive self-promotion or impartiality, which is frowned upon in scientific research, and thus must be refrained from as much as possible.

§  Without the details of the authors, I cannot judge this point.

 Thursday, 28 March 2023

Author Response

Dear Reviewer 3,

Thank you for taking the time to read and comment on this article. I have gone through and made many of the suggested changes. Please find attached the new version with your report copied on at the end and my comments or questions about this--I am sorry for this sloppy format, I wanted to submit to documents but the website did not allow me to do so.

Thank you again for your guidance and advice!

Kind Regards,

Author

Round 2

Reviewer 3 Report

The required has been done. 

Author Response

Thank you for taking the time to read the updated version of the article and for approving the changes.

Best regards,

Eve